# The Effects of Inclisiran on the Subclinical Inflammatory Markers of Atherosclerotic Cardiovascular Disease in Patients at High Cardiovascular Risk

**DOI:** 10.3390/ph18060832

**Published:** 2025-06-01

**Authors:** Mateusz Maligłówka, Adrianna Dec, Łukasz Bułdak, Bogusław Okopień

**Affiliations:** Department of Internal Medicine and Clinical Pharmacology, School of Medicine in Katowice, Medical University of Silesia in Katowice, 40-007 Katowice, Polandlbuldak@gmail.com (Ł.B.);

**Keywords:** inclisiran, heterozygous familial hypercholesterolemia, pentraxin-3, interleukin-18, sCD40L

## Abstract

**Background/Objectives:** Hypercholesterolemia, accompanied by vascular inflammation, leads to the premature initiation and progression of atherosclerosis, and both are considered nowadays as well-established cardiovascular (CV) risk factors. For several years, proprotein convertase subtilisin/kexin type 9 inhibitors (PCSK9is), drugs that reduce the degradation of the receptors for low-density lipoprotein cholesterol (LDLRs), have appeared to be a very efficient lipid-lowering therapy among patients with complications resulting from atherosclerotic cardiovascular disease (ASCVD). Previous studies showed that drugs used to fight hypercholesterolemia (predominantly statins) have significant pleiotropic effects, including anti-inflammatory effects. To date, data on the potential impact of PCSK9 inhibitors, especially inclisiran, on the course of inflammation is still lacking. Therefore, we conceived a study to evaluate the effects of inclisiran on the markers of subclinical inflammation (e.g., pentraxin 3 (PTX3), interleukin-18 (IL-18), and soluble cluster of differentiation 40 ligand (CD40L)) and compared their magnitude in patients at high CV risk, with and without established heterozygous familial hypercholesterolemia (HeFH). **Methods**: A total of 24 patients at high cardiovascular risk, according to European Society of Cardiology (ESC) guidelines, with or without concomitant HeFH diagnosed using Dutch Lipid Clinic Network (DLCN) criteria, were enrolled in this study. Lipid concentrations and levels of subclinical inflammatory markers of atherosclerosis were measured at the beginning and after 3 months of therapy. **Results**: After three months of therapy with inclisiran, a statistically significant reduction included **total cholesterol (TC): study group 1:** from 287.6 ± 94.15 to 215.2 ± 89.08 [mg/dL], *p* = 0.022 and **study group 2:** from 211.71 ± 52.72 to 147.64 ± 55.44 [mg/dL], *p* < 0.001, and **low-density lipoprotein cholesterol (LDL-c): study group 1:** from 180.79 ± 73.33 to 114.65 ± 71.54 [mg/dL], *p* = 0.031 and **study group 2:** from 129.62 ± 46.75 to 63.39 ± 43.6 [mg/dL], *p* < 0.001. Moreover significant drops were observed in concentrations of **PTX3: study group 1:** from 1336.33 ± 395.15 to 1121.75 ± 351.17 [pg/mL], *p* = 0.013 and **study group 2:** from 1610.76 ± 537.78 to 1376.92 ± 529.19 [pg/mL], *p* = 0.017), and **IL-18: study group 1**: from 11.89 (9.72–13.98) to 9.15 (8.62–10.06) [pg/mL], *p* = 0.005 and **study group 2**: from 11.58 (10.87–16.97) to 9.65 (8.43–10.95) [pg/mL], *p* = 0.003). There were no significant changes in the levels of sCD40L. **Conclusions**: This study confirmed the ability of inclisiran to reduce LDL-c levels in patients at high cardiovascular risk just after one dose of the drug. Furthermore, it appeared that beyond its lipid-lowering effect, the drug may also affect some inflammatory processes involved in the initiation and progression of atherosclerosis.

## 1. Introduction

Complications resulting from atherosclerosis, such as myocardial infarction or stroke, are still the most frequent causes of death worldwide [1]. Taking into consideration one of the strongest and widely proven positive correlations between hypercholesterolemia and the incidence of cardiovascular events, the number of patients who constantly do not meet target goals concerning low-density cholesterol (LDL-c) level in the blood still haunts public healthcare systems [2,3].

Among patients with abnormal lipid profiles and increased cardiovascular risk, there are patients with heterozygous familial hypercholesterolemia (HeFH). The relatively common genetic disorder, with an estimated prevalence in the population of 1 in 200 to 250 individuals, results in elevated levels of LDL-c and leads to the premature atherosclerotic plaque formation [4]. Significant hypercholesterolemia, in combination with increased levels of inflammatory biomarkers of atherosclerosis described in patients with HeFH, such as hs-CRP (high sensitivity C-reactive protein), causes subsequent cardiovascular risk elevation [5,6,7,8]. The early initiation of intensive lipid-lowering therapy in patients with HeFH is necessary to reduce the incidence of cardiovascular complications [9].

Another group of patients who require intensive lipid-lowering therapy are individuals at high cardiovascular risk, often with numerous comorbidities and a history of cardiovascular events [10]. Unfortunately, in view of the significantly elevated LDL-c and strict treatment goal, the results of combined statin and ezetimibe treatment remain disappointing, and patients often require additional lipid-lowering therapy, including drugs that act on the proprotein convertase subtilisin/kexin type 9 (PCSK9) axis in LDL-c reduction [11,12].

For several years, proprotein convertase subtilisin/kexin type 9 inhibitors (PCSK9is), drugs that reduce the degradation of the receptors for low-density lipoprotein cholesterol (LDLRs), have appeared to be a very efficient lipid-lowering therapy among patients with complications resulting from atherosclerotic cardiovascular disease (ASCVD). Fully humanized monoclonal antibodies used against the circulating form of PCSK9 (PCSK9 mAbs) (e.g., alirocumab and evolocumab) and small-interfering RNA (siRNA), which inhibits the translation of PCSK9 mRNA (inclisiran), have allowed the achievement of LDL-c treatment goals in the groups of patients at high cardiovascular risk [13,14,15].

Inclisiran, as a siRNA, silences the translation of the PCSK9 gene transcript and thus prevents the formation of mature protein. The biochemical structure, enriched with a specific ligand—triantenary N-acetylgalactosamine (GalNAc), limits the action of the drug only to the liver cells. Furthermore, inclisiran, being metabolized by non-specific nucleases and not affecting cytochrome P450, has no drug interactions [16]. In RCTs (ORION-9, ORION-10, and ORION-11), the drug caused LDL-c reduction by 40 to 51% from the baseline with only two subcutaneous injections per year and without any serious adverse effects. To date, there is no data from RCTs concerning their influence on cardiovascular risk reduction [16,17,18,19].

Current lipid-lowering therapies, in majority based on oral drugs (e.g., statins monotherapy or in combination with ezetimibe), have turned out to be very effective in the reduction in LDL-c levels, which has been followed by a lower incidence of cardiovascular events and mortality rate [20,21,22]. This beneficial effect was observed mainly in secondary prevention and is considered as partially attributed to the pleiotropic effects affecting other extralipid pathogenetic factors in atherosclerosis initiation and progression (e.g., inflammatory status, oxidative stress, plaque stabilization, and platelet and endothelial function) [23].

The role of inflammation in the development and progression of atherosclerosis has been unequivocally confirmed [24,25,26]. Furthermore, anti-inflammatory drugs (e.g., colchicine or canakinumab) appeared to significantly lower the risk of ischemic cardiovascular events in secondary prevention in patients with prior myocardial infarction (COLCOT trial) [27,28]. The involvement of pentraxin-3 (PTX3) in the pathophysiology of atherosclerosis especially concerns the modulation of the vascular inflammatory response and enhances endothelial cell damage [29,30]. Interleukin-18 (IL-18), as a proinflammatory cytokine, mediates proatherogenic effects and is involved in the process of atherosclerotic plaque destruction [31]. A cluster of differentiation 40 (CD40) and CD40 ligand (CD40L, costimulatory molecules that belong to the superfamily of tumor necrosis factor (TNF) proinflammatory chemokines are contributory to the burden of atherosclerosis, plaque stability, and prothrombotic effects [32,33,34].

The multifaceted etiology of atherosclerosis, the pleiotropic effects of the current oral lipid-lowering drugs, and the still not fully elucidated impact of inclisiran on the different pathogenetic factors in atherosclerosis have led us to explore its influence on the subclinical inflammatory markers of the disease i.e., PTX3, IL-18, and the soluble cluster of differentiation 40 ligand (sCD40L). Therefore, we conceived a study to compare the influence of a 3 month treatment with inclisiran on laboratory parameters involved in the pathophysiology of atherosclerosis in patients at high cardiovascular risk, with and without concomitant HeFH, to determine the possible differences in response to the therapy depending on the etiology of atherosclerosis. Furthermore, it was conducted with the purpose of searching for the pleiotropic effects of inclisiran.

## 2. Results

This study included a total of 24 patients: 10 patients at high cardiovascular risk diagnosed with HeFH (study group 1) and 14 patients at very high cardiovascular risk without concomitant HeFH (study group 2). The mean age of the patients was statistically lower in study group 1: 54 years old (±10 y. o.) vs. study group 2: 63 years old (±10 y. o.) (*p* = 0.032). Furthermore, at the beginning of this study, there was a statistically higher level of TC in study group 1: 287.6 ± 94.1 vs. study group 2: 211.7 ± 52.7 [mg/dL] (*p* = 0.038). The difference in LDL-c between the groups seemed relatively high (study group 1: 180.79 ± 73.33 vs. study group 2: 129.62 ± 46.75 [mg/dL]), but the statistical significance has not been reached (*p* = 0.072). The fasting blood glucose was lower in study group 1: 93.9 (89–95.3) vs. study group 2: 102 (98.5–106) [mg/dL] (*p* = 0.0091), which was associated with a nearly statistically significant difference in the glycated hemoglobin (HbA1c) level: study group 1: 5.58 (5.18–5.7) vs. study group 2: 5.95 (5.61–6.06) [%] (*p* = 0.053).

At the beginning of this study, there were no statistically significant differences in the levels of subclinical inflammatory biomarkers between the study groups. The detailed baseline characteristics of the patients are presented in Table 1.

After three months of therapy with inclisiran, among basic blood parameters, a statistically significant reduction in TC (study group 1: from 287.6 ± 94.15 to 215.2 ± 89.08 [mg/dL], *p* = 0.022; study group 2: from 211.71 ± 52.72 to 147.64 ± 55.44 [mg/dL], *p* < 0.001) (Figure 1) and LDL-c (study group 1: from 180.79 ± 73.33 to 114.65 ± 71.54 [mg/dL], *p* = 0.031; study group 2: from 129.62 ± 46.75 to 63.39 ± 43.6 [mg/dL], *p* < 0.001) (Figure 2) were observed in both groups. Nevertheless, there were no statistical differences in the magnitude of the reduction in study group 1: ΔTC = −72.4 ± 82.79 [mg/dL]; study group 2: ΔTC = −64.07 ± 33.84 [mg/dL], *p* = 0.769; study group 1 ΔLDL-c = −66.14 ± 81.86 [mg/dL]; and study group 2: ΔLDL-c = −66.23 ± 27.05 [mg/dL], *p* = 0.997). A total of 10% of patients from study group 1 and 42.9% of patients from study group 2 achieved a target level of LDL-c below 70 mg/dL (according to ESC guidelines’ target level of LDL-c for patients at high cardiovascular risk), nonetheless with no statistically significant difference between two groups, *p* = 0.2416.

In the group of the subclinical inflammatory markers of atherosclerosis assessed in this study, significant drops in both groups concerned PTX3 (study group 1: from 1336.33 ± 395.15 to 1121.75 ± 351.17 [pg/mL], *p* = 0.013; study group 2: from 1610.76 ± 537.78 to 1376.92 ± 529.19 [pg/mL], *p* = 0.017) (Figure 3), and IL-18 (study group 1: from 11.89 (9.72–13.98) to 9.15 (8.62–10.06) [pg/mL], *p* = 0.005; study group 2: from 11.58 (10.87–16.97) to 9.65 (8.43–10.95) [pg/mL], *p* = 0.003) (Figure 4). However, no differences between the groups in the extent of the reduction (study group 1: ΔPTX3 = −214.58 ± 218.69 [pg/mL], study group 2 ΔPTX3 = −233.84 ± 320.62 [pg/mL], *p* = 0.8627; study group 1: ΔIL-18 = −2.94 (−4.65−(−0.62)) [pg/mL]; study group 2: ΔIL-18 = −2.17 (−6.77–(−1.34)) [pg/mL], *p* = 0.3564) were seen.

Contrary to previously described markers, there were no significant changes in the level of sCD40L (study group 1: from 4.37 ± 2.56 to 5.29 ± 2.45 [ng/mL], *p* = 0.3897; study group 2: from 3.79 ± 2.08 to 4.7 ± 1.95 [ng/mL], *p* = 0.7986).

During the course of this study, there were no serious or minor adverse effects.

## 3. Discussion

The statistical differences in the initial assessment of the study groups concerned three parameters: age, concentration of total cholesterol, and fasting glucose level. The patients at high cardiovascular risk from study group 1 were statistically younger than the patients from study group 2, which most probably is a result of their HeFH diagnosis. Regardless of their prior history of cardiovascular events or age, according to ESC guidelines for the management of dyslipidemias, they have to be assigned to the high cardiovascular risk group [15]. Furthermore, genetic alterations in the lipid metabolism pathways in the patients with HeFH seemed to be responsible for the statistically higher concentration of cholesterol in study group 1 in comparison with study group 2 [4]. The differences in fasting glucose levels might result from those concerning the age between two groups. The age is considered to be a risk factor for glucose metabolism impairment, which reflects the statistically higher fasting glucose level in study group 2 [35].

After 3 months of therapy, inclisiran appeared to be effective in lowering the TC and LDL-c plasma levels in both groups to the extent that was similar to changes observed in prior RCTs (e.g., ORION−3, ORION 9–11) in patients at high cardiovascular risk, regardless of concomitant HeFH [18,19,36].

The analysis of the impact of inclisiran on the subclinical inflammatory markers of atherosclerosis showed a statistically significant reduction in PTX3 in both groups. PTX3, also known as TGS-14 (tumor necrosis factor—stimulated gene 14), is a long-chain glycoprotein belonging to the innate immune system and functions as a rapid indicator of early inflammatory processes, i.e., through the regulation of the complement system. It is produced by endothelial cells, vascular smooth muscle cells, neutrophils, and macrophages and can be involved in the development of atherosclerosis by intensifying endothelial dysfunction, interfering with angiogenesis, and increasing the amount of reactive oxygen species (ROS) [37]. Elevated levels of PTX3 were observed both in mice with atherosclerosis as well as in patients with acute coronary syndromes, predicting poor prognosis [38,39,40]. Current data shows that PTX3 could become a useful predictive biomarker in cardiovascular diseases or even a potential therapeutic target [37]. Data from this study suggest that inclisiran may affect the subclinical inflammatory status of patients with atherosclerosis, irrespective of the etiology of the disease. Further investigations are necessary to fully elucidate the involvement of PTX3 in the pathophysiology of atherosclerosis and the influence of different PCSK9 inhibitors on PTX3 plasma levels.

A significant decrease in IL-18 levels was observed in both study groups after 3 months. IL-18, a proinflammatory cytokine from the superfamily of interleukin 1 (IL-1), which is responsible for regulating immune response, was also described as aggravating atherosclerosis [41,42]. Secreted mainly from macrophages, IL-18 was related to the atherosclerotic plaque destruction through the induction of interferon gamma (IFN-γ) in T cells and natural killer (NKs) cells [43]. Furthermore, its serum levels were described as strong predictors of death in patients with stable and unstable angina pectoris [44]. Data concerning the influence of anti-PCSK9 drugs on IL-18 plasma levels is limited to one small study in humans treated with alirocumab, with an observed significant decrease in proinflammatory cytokine levels, i.e., IL-18 (*p* < 0.01) [45]. The coherent observations presented in the aforementioned study and this study allow us to assume the positive influence of drugs limiting the action of PCSK9, and also in reducing inflammation in patients with atherosclerosis. Nevertheless, further research is necessary to confirm this.

The interaction between CD40 and CD40L, cytokines involved in the modulation of the immune response that belong to the TNF receptor and its ligand superfamily, has appeared to be correlated with the occurrence and progression of atherosclerosis [46]. The inhibition of CD40L co-signaling in mice, especially concerning CD4+ T lymphocytes and macrophages, was related to plaque area reduction and increased plaque stability, with no negative effects on immune response [47]. Furthermore, soluble CD40L (sCD40L), a fragment of the transmembrane glycoprotein that is released into the bloodstream, intensifies the progression of atherosclerosis by promoting prothrombotic effects, and it is correlated with lower levels of high-density cholesterol (HDL-c) [48]. Significantly higher levels of sCD40L were observed in patients with stable coronary heart disease (*p* < 0.05), as well as in patients with acute coronary syndromes (*p* < 0.01) [49,50]. Nevertheless, in this study, inclisiran appeared to have no statistically significant influence on the level of sCD40L in either study group. Further research is essential to elucidate the mechanisms responsible for a lack of change in the sCD40L’s concentration. However, all the above findings may suggest that, being an effective lipid-lowering drug, inclisiran does not affect all types of inflammatory responses, which could result in a lower risk of immunosuppression, especially in the context of long-term therapy.

Some limitations should be kept in mind while analyzing the results of this study. A small group of patients was analyzed in a short period of time. It was a single-center study that concerned a local population of Caucasians, and findings cannot be generalized to different races. Due to ethical issues, there was no control group—all patients with indications for lipid-lowering therapy escalation were treated with inclisiran. The number of patients with HeFH may be underestimated due to the reliance on clinical criteria alone.

## 4. Materials and Methods

This study was conducted between June 2023 and November 2024 in the Outpatient Department for the Treatment of Metabolic Diseases, belonging to the Internal Medicine and Clinical Pharmacology Ward at the University Clinical Center of Prof. K. Gibiński of the Medical University of Silesia in Katowice. A total of 24 patients, aged 41–81 years old with at least high cardiovascular risk, who did not achieve the LDLC plasma target level according to the 2021 ESC Guidelines on cardiovascular disease prevention in clinical practice, met the inclusion criteria, and did not meet the exclusion criteria, were enrolled in this study. There were 13 women and 11 men. A total of 10 patients (study group 1) were diagnosed with HeFH (scoring at least 9 points, according to Dutch Lipid Clinic Network (DLCN) criteria), and the latter 14 patients (study group 2) had very high cardiovascular risk with no concomitant HeFH (scoring less than 9 points according to DLCN criteria). Subclinical inflammatory markers of atherosclerosis were measured at the beginning and after three months of therapy with inclisiran in both groups. Each patient provided written informed consent in accordance with the Helsinki Declaration. The Medical University of Silesia’s Bioethical Committee accepted the study protocol (BNW/NWN/0052/KB1/53/23). Figure 5 presents a study flow chart.

### 4.1. Study Design

The inclusion criteria were as follows: age over 18 years old, obtained written informed consent, concomitant hypercholesterolemia with at least high cardiovascular risk and exceeded therapeutic goal (LDL-c over the required level characteristic to the specific cardiovascular risk groups according to ESC guidelines), prior statin treatment at high doses for at least 3 months (rosuvastatin 20–40 mg per day or atorvastatin 40–80 mg per day), in case of statin intolerance, prior statin treatment at highest tolerated dose for at least 3 months, prior treatment with ezetimibe for at least 1 month, alanine aminotransferase (ALT) or aspartate aminotransferase (AST) < 3 upper limit level, creatine kinase (CK) < 5 upper limit level, and TSH (thyroid-stimulating hormone) level at reference range.

The exclusion criteria were as follows: age under 18 years old, secondary hypercholesterolemia, decompensated heart failure (grade III–IV according to the New York Heart Association (NYHA)), diabetes mellitus (DM), history of oncological treatment within 5 years, liver failure (class B–C according to Child–Pugh classification), acute and chronic inflammatory diseases, poor compliance, lack of informed consent, pregnancy or breastfeeding, history of hypersensitivity reactions to inclisiran, and laboratory disturbances as follows: ALT or AST > 3 upper limit level or bilirubin > 1.2 mg/dL, hemoglobin < 10 g/dL or >17 g/dL, red blood cell count (RBC) < 3.5 M/μL or >5.5 M/μL, white blood cell count (WBC) < 3.5 K/μL or >10 K/μL, platelet count < 140 K/μL or >400 K/μL, and triglycerides (TG) > 500 mg/dL.

Laboratory blood tests included the following: complete blood count, creatinine with estimated glomerular filtration rate (eGFR), markers of liver function [bilirubin, AST, ALT, and international normalized ratio (INR)], glucose, HbA1c, CK, lipid profile [TC, LDL-c, HDL-c, TG], TSH, N-terminal prohormone B-type natriuretic peptide (NT-proBNP), and the subclinical inflammatory markers of atherosclerosis: pentraxin 3 (PTX3), CD40L and interleukin 18 (IL-18). They were measured at the beginning and after three months of therapy.

Basic laboratory analyses were performed using commercially available equipment (Sysmex XN1000, Sysmex Corp., Kobe, Japan; ACL 500, Instrumentation Laboratory, Bedford, MA, USA; RAPIDPoint 500, Siemens Healthineers, Erlangen, Germany; Cobas PRO,, Roche Diagnostics, Basel, Switzerland). The analyses of the subclinical markers of atherosclerosis were performed with the use of respective ELISA detection kits (HUMAN PENTRAXIN 3 ELISA, BioVendor R&D, Brno, Czech Republic; Enzyme-linked immunoabsorbent Assay Kit for IL-18, Cloud-Clone Corp. Houston, TX, USA; Human sCD40L ELISA, Diaclone SAS, Besancon, France) and an xMark Microplate Absorbance Spectrophotometer (Bio-Rad Laboratories, Inc., Hercules, CA, USA).

### 4.2. Statistical Analyses

The necessary sample size was estimated using previous experimental data on the impact of pitavastatin on PTX-3 levels. In patients with elevated PTX-3, pitavastatin was shown to reduce the PTX-3 concentration by around 20% during three-month therapy [51]. This was translated to show that the sample necessary to provide statistically meaningful results should not be less than 10 [with type I (α) error set at 0.05 and statistical power set at 80%]. All data were collated in a Microsoft Excel 365 spreadsheet (Microsoft Corp., Redmond, WA, USA) and transferred to the Statistica software package v. 13.0 (StatSoft Inc., Tulsa, OK, USA) and Plus Set v. 5.0 (TIBCO Software Inc., Palo Alto, CA, USA). The normality of distribution was evaluated using the Shapiro–Wilk’s test. The data is presented in tables as the mean with standard deviation (SD) for a normal distribution or median with interquartile range (IQR) for a non-normal distribution. The baseline statistical differences and the differences in response to the therapy with inclisiran between the two study groups were assessed with Welch’s *t*-test (means for normal distribution) and the Mann–Whitney test (medians for non-normal distribution). The statistical differences between the continuous data at the beginning and after the therapy with inclisiran were assessed with the Student’s *t*-test (means for normal distribution) and the Wilcoxon (medians for non-normal distribution). A value of *p* < 0.05 indicated statistical significance.

## 5. Conclusions

This study confirmed the ability of inclisiran to reduce LDLC levels in patients at high cardiovascular risk after just one dose of the drug. Furthermore, it appeared that beyond the lipid pathway of atherosclerosis, the drug may also affect some inflammatory processes involved in the initiation and progression of the disease. Nonetheless, there is a constant need for further research, including RCTs, that precisely and unequivocally find out the influence of inclisiran on the different pathogenetic factors of atherosclerosis. It may contribute to a reduction in the number of deaths resulting from atherosclerotic complications.

## Figures and Tables

**Figure 1 pharmaceuticals-18-00832-f001:**
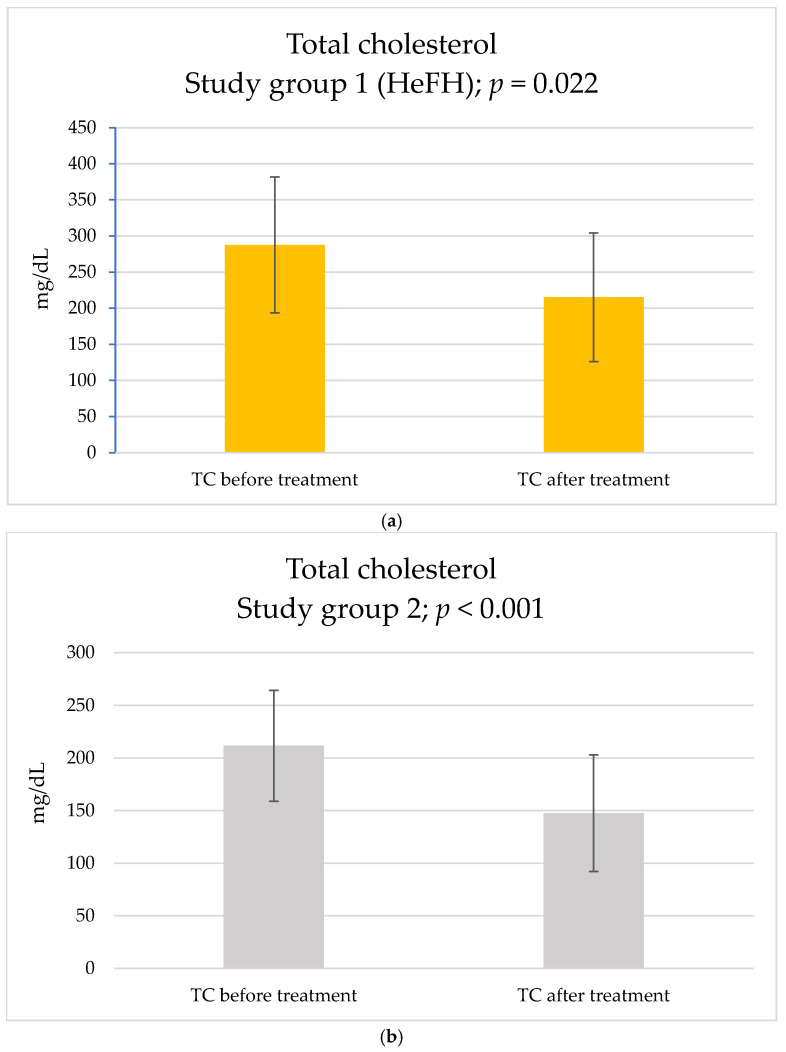
Effect of inclisiran on TC: (**a**) study group 1 and (**b**) study group 2.

**Figure 2 pharmaceuticals-18-00832-f002:**
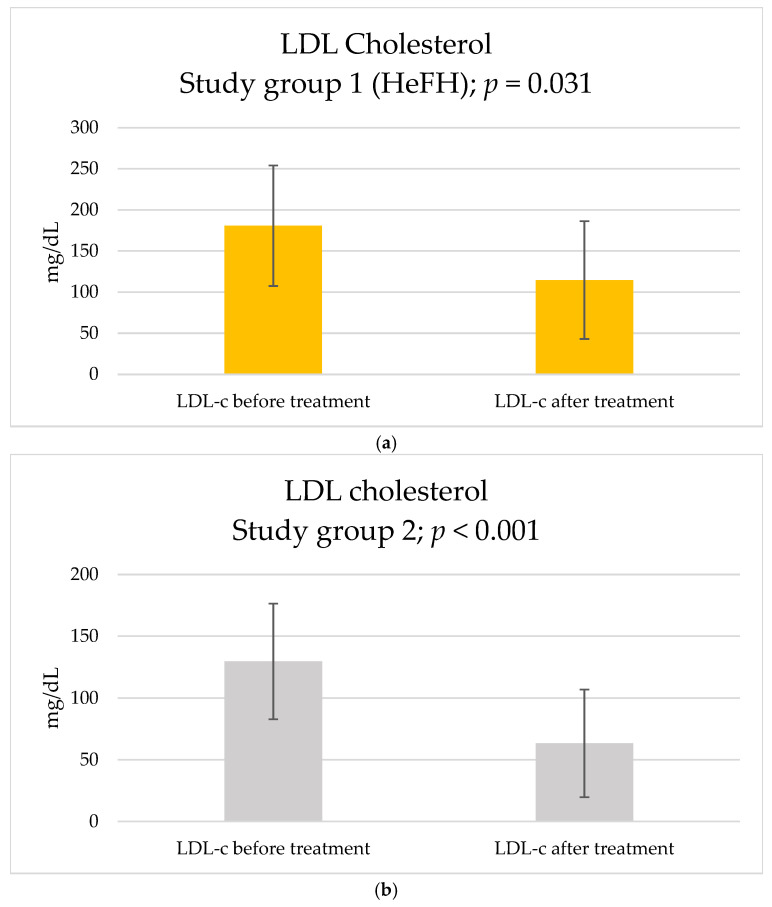
Effect of inclisiran on LDL-c: (**a**) study group 1 and (**b**) study group 2.

**Figure 3 pharmaceuticals-18-00832-f003:**
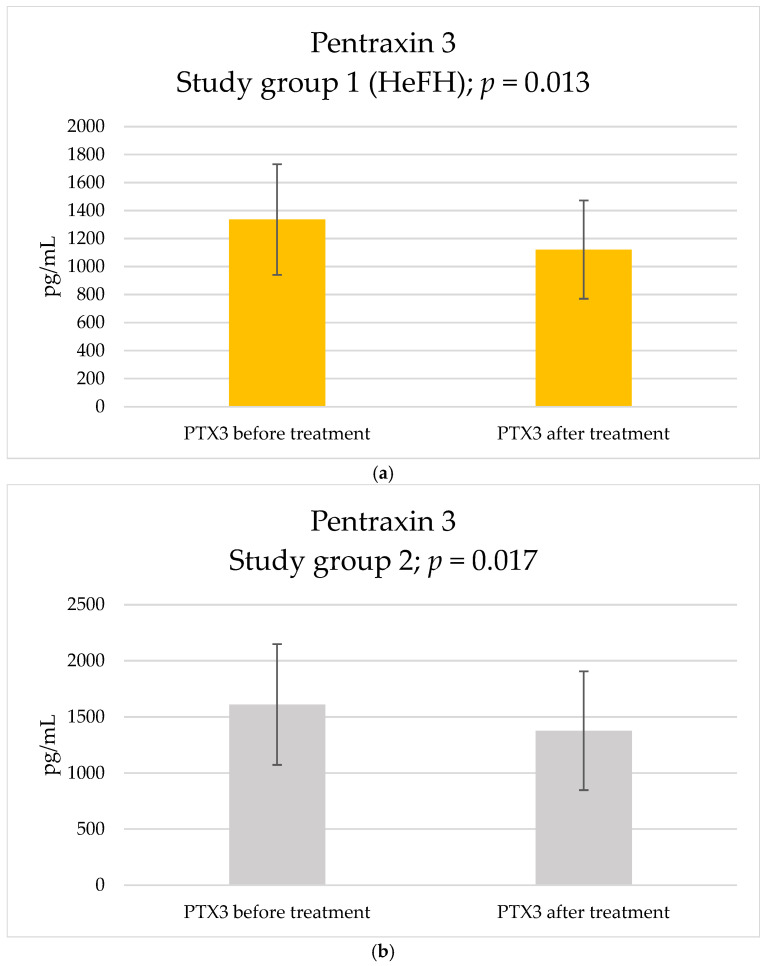
Effect of inclisiran on PTX-3 concentration: (**a**) study group 1 and (**b**) study group 2.

**Figure 4 pharmaceuticals-18-00832-f004:**
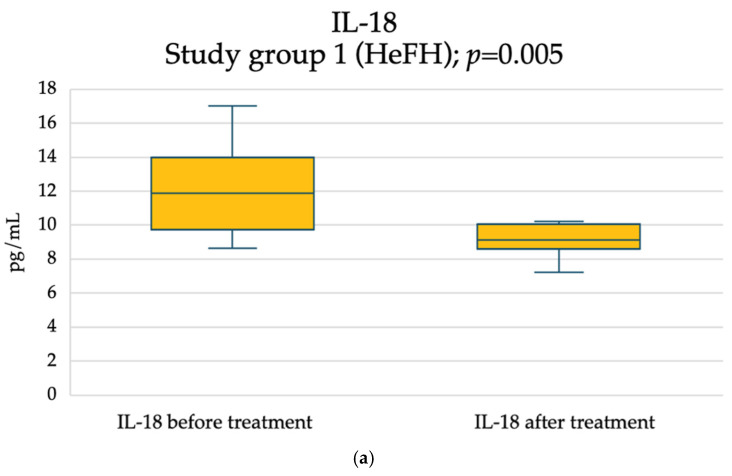
Effect of inclisiran on IL-18 concentration: (**a**) study group 1 and (**b**) study group 2.

**Figure 5 pharmaceuticals-18-00832-f005:**
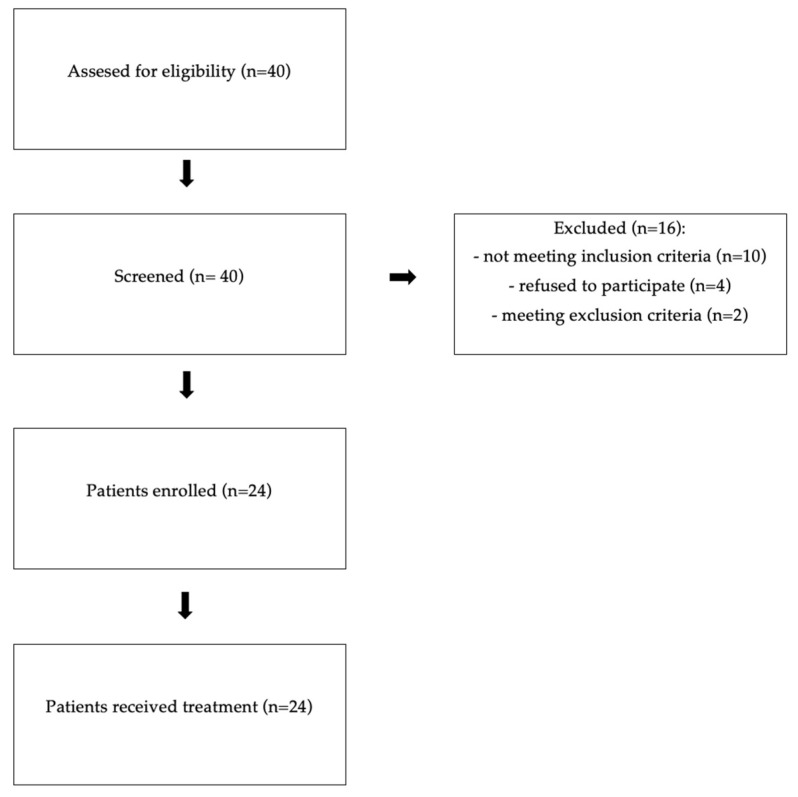
Study flow chart.

**Table 1 pharmaceuticals-18-00832-t001:** Baseline characteristics of study groups.

	Study Group 1 (HeFH); n = 10	Study Group 2; n = 14	Statistical Test	*p* Value
**Age, years**	54 ± 10	63 ± 10	Welch’s *t*-test	**0.032**
**WBC, K/μL**	5.87 ± 1.39	6.4 ± 1.06		0.333
**RBC, M/μL**	4.57 ± 0.43	4.78 ± 0.56		0.333
**HGB, g/dL**	13.73 ± 1.48	14.69 ± 1.37		0.126
**PLT, K/μL**	247.6 ± 68.89	247.92 ± 42.38		0.99
**TC, mg/dL**	287.6 ± 94.1	211.7 ± 52.7		**0.038**
**LDL-c, mg/dL**	180.8 ± 73.3	129.6 ± 46.8		0.072
**HDL-c, mg/dL**	57.5 (51.2–121.75)	47.6 (40.95–59.63)	M-W	0.064
**TG, mg/dL**	117.3 ± 50.8	144.9 ± 50.8	Welch’s *t*-test	0.206
**Total bilirubin, mg/dL**	0.46 ± 0.18	0.71 ± 0.3		0.043
**ALT, U/L**	26 ± 13.8	33 ± 16.9		0.28
**AST, U/L**	24.7 (23.1–26.7)	26 (19–31.1)	M-W	0.886
**INR**	0.92 (0.87–1)	0.92 (0.88–1.02)		0.761
**Creatinine, mg/dL**	0.67 (0.64–0.8)	0.86 (0.76–0.97)		0.053
**CK, U/L**	152.5 (128.5–183.3)	122.5 (66.5–166.3)		0.187
**Glucose, mg/dL**	93.9 (89–95.3)	102 (98.5–106)		**0.009**
**HbA1c, %**	5.58 (5.18–5.7)	5.95 (5.61–6.06)		0.053
**TSH, uIU/mL**	1.76 (1.41–2.88)	1.54 (0.81–2.88)		0.656
**NT-proBNP, pg/mL**	95.35 (35.15–154.75)	81 (39.48–183)		0.721
**PTX3, pg/mL**	1336.33 ± 395.15	1610.76 ± 537.78	Welch’s *t*-test	0.164
**IL-18, pg/mL**	11.89 (9.72–13.98)	11.58 (10.87–16.97)	M-W	0.558
**sCD40L, ng/mL**	4.37 ± 2.56	3.79 ± 2.08	Welch’s *t*-test	0.563

Abbreviations: ALT—alanine aminotransferase; AST—aspartate aminotransferase; sCD40L—soluble CD40 ligand; CK—creatine kinase; HbA1c—glycated hemoglobin; HDL-c—high-density lipoprotein cholesterol concentration; HGB—haemoglobin; IL-18—interleukin 18; INR—international normalized ratio; LDL-c—low-density lipoprotein cholesterol concentration; M-W—Mann–Whitney test; NT-proBNP—N-terminal pro B-type natriuretic peptide; PLT—platelet count; PTX3—pentraxin 3; RBC—red blood cell count; TC—total cholesterol; TG—triglycerides; TSH—thyroid-stimulating hormone; WBC—white blood cell count.

## Data Availability

The original contributions presented in this study are included in the article. Further inquiries can be directed to the corresponding author.

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
