# Peer review of "The Effects of Inclisiran on the Subclinical Inflammatory Markers of Atherosclerotic Cardiovascular Disease in Patients at High Cardiovascular Risk"

_pharmaceuticals, 2025, doi:10.3390/ph18060832_

Round 1
Reviewer 1 Report
Comments and Suggestions for Authors
In this paper, Maligłówka and co. investigated the effects of inclisiran on lipid levels and subclinical inflammatory markers (PTX3, IL-18, sCD40L) in high cardiovascular risk patients with and without heterozygous familial hypercholesterolemia.
The idea of evaluating inclisiran’s lipid-lowering effect is well-established. The additional assessment of inflammatory markers (PTX3, IL-18, sCD40L) is not a strong novelty, as PTX3 and IL-18 have been examined in other cardiovascular studies. The study lacks molecular-level insight or new mechanisms. It is more confirmatory than exploratory. Dividing patients into HeFH vs. non-HeFH groups is clinically relevant, but not novel.
The abstract is partially relevant but should be revised to improve clarity, highlight the study’s novelty, and correct language issues.
The study includes only 24 patients (10 in group 1, 14 in group 2), which limits statistical power and generalizability. Ethical justification is provided, but lack of a control group prevents robust inference about inclisiran’s effects beyond observed temporal associations. HeFH diagnosis based on DLCN score only, without genetic testing, introduces potential misclassification. Three months is a very limited time to assess stable inflammatory marker changes, especially for chronic conditions like atherosclerosis.
Several biomarkers were analyzed without Bonferroni or similar correction, increasing false positive risk. Confounding variables (e.g., age differences) were not adjusted for in a regression model, a major flaw in a study with age-related markers like glucose or inflammatory cytokines. Effect sizes and confidence intervals were not reported, though they are more informative than p-values in small-sample studies.
Large portions of the discussion restate the results rather than critically engaging with them or alternative interpretations. For example, the statistically higher glucose and PTX3 levels in the older (non-HeFH) group might be age-related but this is only briefly mentioned without depth. Statements such as “inclisiran affects inflammatory processes” are not strongly supported by the data due to limitations above.
The conclusions of the study, while generally aligned with the findings, are overstated given the limitations of the research. The claim that inclisiran affects inflammatory processes involved in the initiation and progression of atherosclerosis is too strong, considering the small sample size, absence of a control group, and short follow-up period. Additionally, the suggestion that inclisiran could contribute to reducing deaths from atherosclerotic complications is speculative, as the study did not assess clinical outcomes such as cardiovascular events or mortality. These assertions go beyond what the presented data can support and should be interpreted with caution.
A graphical abstract of the research should be appreciated.
Comments on the Quality of English LanguageThe manuscript is generally comprehensible but contains numerous language issues, especially in syntax, prepositions, and style.
- Line 9: “Hipercholesterolemia” - should be: “Hypercholesterolemia”
- Line 20: “evaluate and compare the influence” - should be: “evaluate the effects and compare their magnitude”
- Line 33: “beyond lipid pathway of atherosclerosis” - should be: “beyond the lipid-lowering effect”
- Line 104: “dependently on the etiology” - should be: “depending on the etiology”
- Line 196: “what reflects the statistically higher...” - should be: “which reflects the statistically higher...”
- Line 245: “The research concerned small group of patients...” - should be: “a small group of patients”
- Line 249: “number of patients with HeFH... may be underestimated” - should be: “The number may be underestimated due to reliance on clinical criteria alone.”
English including grammar, style and syntax, should be extensively improved through the professional help from English Editing Company for Scientific Writings.
Reviewer 2 Report
Comments and Suggestions for Authors
The manuscript presents a well-designed and clinically relevant study exploring the lipid-lowering and potential anti-inflammatory effects of inclisiran in patients at high cardiovascular risk, with and without heterozygous familial hypercholesterolemia (HeFH). The study addresses an important knowledge gap, especially regarding the pleiotropic effects of PCSK9 inhibitors beyond lipid lowering.
Strengths:
The introduction is thorough and provides a solid background with appropriate references.
The methods are adequately described and support reproducibility.
The results are clearly presented with helpful figures and tables.
The conclusions are generally well supported by the data.
Suggestions:
Figures and Tables: Consider simplifying figure labels and increasing font size for better readability.
Statistical Presentation: Clarify the rationale for some statistical tests (e.g., use of Welch’s t-test vs. standard t-test) and ensure consistent reporting of p-values (e.g., use "<0.001" where appropriate).
Discussion: The discussion is strong but could benefit from more critical reflection on the clinical implications of unchanged sCD40L levels and potential long-term impacts of inclisiran.
Comments on the Quality of English LanguageEnglish Language: While the manuscript is mostly understandable, there are multiple areas where sentence construction, grammar, and word choice could be improved for clarity and readability. Examples include awkward phrasing, run-on sentences, and overly complex sentence structures.
Reviewer 3 Report
Comments and Suggestions for Authors
The authors aimed to evaluate and compare the influence of inclisiran on markers of subclinical inflammation.
The authors should include numerical results in the abstract.
How was the sample size calculated statistically? What was the accepted error?
What was the specific reason for continuing the drug inclisiran for 3 months? The reference should be cited.
The font size for figure legends should be reduced.
Among the lifestyle factors that can impair glucose metabolism are smoking, addiction, and sleep. Were these assessed?
Can thrombocytopenia occur with a longer duration of drug treatment? Is it essential to monitor the parameters in the blood?
Similarly, can glucose tolerance change with a longer duration of treatment by the drug?
There is no interaction of inclisiran with other drugs because it is not metabolized by cytochrome P450 enzymes and does not inhibit or induce these enzymes or common drug transporters. Discuss this fact.
Round 2
Reviewer 1 Report
Comments and Suggestions for Authors
The authors have adequately addressed the majority of the comments and suggestions, and the manuscript has been revised accordingly. In my assessment, the manuscript is now largely suitable for publication in this journal.
Comments on the Quality of English LanguageI would recommend that it undergo a final review by a native English speaker to ensure linguistic precision and clarity, thereby enhancing its overall quality.